# Compilation of reported protein changes in the brain in Alzheimer's disease

Manor Askenazi [1], Tomas Kavanagh[2], Geoffrey Pires[3], Beatrix Ueberheide[3,4,5], Thomas Wisniewski [3] & Eleanor Drummond [2,3] ✉

Proteomic studies of human Alzheimer's disease brain tissue have potential to identify protein changes that drive disease, and to identify new drug targets. Here, we analyse 38 published Alzheimer's disease proteomic studies, generating a map of protein changes in human brain tissue across thirteen brain regions, three disease stages (preclinical Alzheimer's disease, mild cognitive impairment, advanced Alzheimer's disease), and proteins enriched in amyloid plaques, neurofibrillary tangles, and cerebral amyloid angiopathy. Our dataset is compiled into a searchable database (NeuroPro). We found 848 proteins were consistently altered in 5 or more studies. Comparison of protein changes in early-stage and advanced Alzheimer's disease revealed proteins associated with synapse, vesicle, and lysosomal pathways show change early in disease, but widespread changes in mitochondrial associated protein expression change are only seen in advanced Alzheimer's disease. Protein changes were similar for brain regions considered vulnerable and regions considered resistant. This resource provides insight into Alzheimer's disease brain protein changes and highlights proteins of interest for further study.

The cause of sporadic Alzheimer's disease (AD) is currently unknown. Mass spectrometry-based proteomic studies of human brain tissue are an excellent way to uncover the disease mechanisms involved in AD. Protein changes are particularly important to study for a disease such as AD because post-translational events such as protein accumulation, aggregation or post-translational modifications of proteins directly mediate disease[1]. This has been particularly highlighted by recent studies reporting a poor correlation between mRNA and protein changes in human AD tissue[2–4], thus emphasising the need for proteomic studies when examining AD pathogenesis and identifying new biomarkers and/or drug targets.

In recent years there has been a large increase in the number of human AD proteomic studies[5–7]. These studies have examined protein differences in AD in a variety of brain regions[2–4,8–29] and tissue fractions (e.g. insoluble, synaptic, membrane or blood vessel-enriched

fractions[30–35]). Additional studies have performed localised proteomics of neuropathological lesions such as amyloid plaques, neurofibrillary tangles (NFTs), and cerebral amyloid angiopathy (CAA)[36–42]. Individually, each of these studies have generated important new insight into AD pathogenesis and have uncovered new potential drug targets and biomarkers for AD. Despite these benefits, each of these studies has been somewhat limited when analysed in isolation by either low sample size, inclusion of a limited number of brain regions, or analysis of only one clinical stage of AD.

We hypothesised that a combined analysis of AD human brain tissue proteomic studies would: (1) identify the highest confidence protein changes in AD; (2) resolve potential concerns about inter-study consistency; (3) provide a more comprehensive analysis of AD-associated protein changes that could be used to answer key outstanding questions about AD pathogenesis. Therefore, the aim of this

[1]Biomedical Hosting LLC, Arlington, MA 02140, USA. [2]Brain and Mind Centre and School of Medical Sciences, University of Sydney, Camperdown, NSW 2050, Australia. [3]Center for Cognitive Neurology, Department of Neurology, Grossman School of Medicine, New York University, New York, NY 10016, USA. [4]Proteomics Laboratory, Division of Advanced Research Technologies, Grossman School of Medicine, New York University, New York, NY 10016, USA. [5]Biochemistry and Molecular Pharmacology, Grossman School of Medicine, New York University, New York, NY 10016, USA. ✉e-mail: Eleanor.drummond@sydney.edu.au

study was to perform a combined analysis of mass spectrometry-based proteomic studies examining human AD brain tissue (inclusive of any brain region, any time point in AD, any tissue fraction) to identify consistent protein changes in AD. Inclusion was restricted to studies of human brain tissue, given concerns that animal or cell models do not reflect the complexity of human disease[26,40,43,44]. We have compiled these data into an online database—NeuroPro—which is a user-friendly resource for the scientific community that details protein changes in three clinical stages of AD (preclinical AD, mild cognitive impairment, and advanced AD), 13 brain regions, and proteins enriched in the three neuropathological hallmarks of AD (amyloid plaques, neurofibrillary tangles, and cerebral amyloid angiopathy). Additionally, we demonstrate the utility of NeuroPro, by using this resource to answer key questions about AD pathogenesis, including identification of the earliest protein changes in AD, protein changes associated with selective vulnerability in AD, and correlation of protein enrichment in neuropathological hallmarks and surrounding tissue.

## Results

### Studies included in NeuroPro

Thirty-eight publications met the inclusion criteria for NeuroPro (Table 1 and Fig. 1). This included 32 studies that identified protein differences in bulk tissue homogenate between AD and controls, four studies that identified the proteome of amyloid plaques, two studies that identified the proteome of NFTs and 1 study that identified the proteome of CAA. Combined, these studies resulted in 59 unique comparisons of AD vs controls in bulk tissue. The number of individual comparisons was higher than the number of included publications due to some studies examining either multiple brain regions or multiple stages of AD, which each counted as a unique comparison in our analysis. Together, these bulk tissue studies enabled the comparison of protein changes in 13 brain regions and protein changes at three clinical stages of AD (preclinical AD, MCI, AD). A high variance in the number of differentially expressed proteins (DEPs) identified in each study was observed, largely reflecting the sample size in each study. In sum, NeuroPro currently contains data for 18,119 reported protein changes in AD human brain tissue, corresponding to 5311 significantly altered proteins in AD (Supplementary Data 1).

### Most consistently identified protein changes in AD

Fifty-four proteins were identified as DEPs in at least 15 different studies (NeuroPro score ≥15; Fig. 2B and Supplementary Data 1). The consistent identification of these proteins as significantly altered in AD across so many studies suggests that these protein changes are the most prevalent in AD human brain tissue. 94% (51/54) of these proteins were consistently altered in the same direction across all comparisons: 29 were consistently increased in AD and 22 were consistently decreased in AD. Only three proteins were inconsistently altered: VCAN, UCHL1 and IDH2. VCAN was predominantly increased in AD bulk tissue studies (11 comparisons) but was decreased in the insoluble brain fraction in MCI and preclinical AD[31] and in the CA4/dentate gyrus region of the hippocampus in AD[12]. IDH2 was predominantly decreased in AD (10 comparisons), but was increased in the insoluble fraction of the frontal cortex in preclinical AD and AD[31] and in the wall of the lateral ventricle in AD[18]. UCHL1 showed a more binary split between studies: it was increased in five studies and decreased in eight studies, and there was no obvious reason for this inconsistency; it did not appear linked to differing expression between vulnerable and resistant brain regions, differences between tissue fractions or AD clinical stage.

The most consistently increased proteins in human AD brain tissue were GFAP, APP, HSPB1, CD44 and CLU. The most consistently decreased proteins in human AD brain tissue were VGF, RPH3A, CORO1A, ACTN2 and HOMER1 (Fig. 2B). These proteins were consistently altered in the same direction across multiple brain regions and often also in preclinical AD and MCI. While some of these most

prevalent protein changes were also enriched in neuropathological lesions (Fig. 2B), this was not always the case, showing that protein enrichment in AD bulk tissue does not necessarily equal enrichment in neuropathological lesions. In addition, there were two instances of proteins that were decreased in AD brain tissue but enriched in neuropathological lesions: VGF and SH3GL1 were both consistently decreased in AD brain tissue, but enriched in plaques and CAA, respectively (Fig. 2B). These differences could be due to sample preparation differences between bulk tissue and neuropathological lesion studies, or it may suggest that VGF and SH3GL1 could have unique roles in AD pathogenesis.

Surprisingly, despite consistent detection as DEPs in many AD proteomics studies, 46% of these top 54 proteins are currently understudied in the AD field (classified as ≤10 previous publications linking a protein to AD; Fig. 2B; Supplementary Data 1). In fact, four of these top 54 proteins are novel to the AD field, with no literature directly linking these proteins to AD; two of these novel proteins were consistently increased in AD (CAPG and PBXIP1) and two were consistently decreased in AD (AP3D1 and SUCLA2). Intriguingly, CAPG is also enriched in both plaques and NFTs, suggesting a potentially important role in pathology. The consistent detection of these proteins as significantly altered in AD proteomic studies warrants future studies examining their role in AD and highlights the power of our combined analysis approach.

### Consistency of bulk tissue proteomic studies

There was remarkable consistency between bulk tissue proteomic studies, particularly given different sample preparation and mass spectrometry methods, different brain regions, different solubility states and fractionation steps, and different clinical disease stages. 848 proteins were identified as DEPs in AD in ≥5 comparisons of bulk tissue (Fig. 2A and Supplementary Data 2). Of these proteins, there was very high consistency between studies for the direction of change: 306 proteins were consistently increased in AD vs controls and 442 proteins were consistently decreased in AD vs controls. Only 100 proteins showed an inconsistent directional change in AD between studies. It is important to note that inconsistency between studies does not necessarily suggest technical issues, but rather could reflect important pathological protein differences between different stages of disease, different brain regions or different tissue fractions.

This subset of 848 proteins altered in ≥5 comparisons were highly interconnected (Fig. 3; $p < 1.0 \times 10^{-16}$; protein–protein interaction enrichment). Within this broader protein network, there was evidence of significant enrichment of proteins associated with particular cellular components or biological processes, many of which are known to be associated with AD. For example, there was evidence of significantly decreased synaptic proteins, mitochondrial proteins and vesicle proteins, while there was evidence of significantly increased extracellular and inflammatory proteins (Fig. 2C, D and Supplementary Data 3). Our analysis suggested that there was a particularly widespread decrease in many structural components and processes associated with synaptic function, including both pre- and post-synaptic proteins, proteins associated with neurotransmitter release (particularly glutamate) and vesicle trafficking (particularly clathrin-coated vesicles). Unexpectedly, our combined analysis also uncovered evidence of decreased kinases associated with tau phosphorylation (including GSK3A, GSK3B, CDK5, MARK1 and ROCK2), despite the increased phosphorylation of tau in AD (Supplementary Data 3). There was also strong evidence of increased blood microparticle proteins in AD, and increased proteins associated with neurofibrillary tangles and amyloid plaques.

Interestingly, this high inter-study consistency suggests that protein changes in AD consistently occur in the same direction, regardless of brain region or tissue fraction examined. The biggest determinant of inconsistency between studies appeared to be the power of individual studies: as expected, lower-powered studies

**Table 1 | Studies included in NeuroPro**

| Reference | Brain region(s) | AD stage(s) | Tissue fraction | Lysis and digestion | DEPs included in NeuroPro | Statistical approach |
|---|---|---|---|---|---|---|
| **Bulk Tissue Homogenate** | | | | | | |
| Andreev et al. (2012)[8] | Temporal cortex | Control, AD | n/a | 8 M urea lysis. Trypsin digestion | 188 | FDR <5% |
| Astillero-Lopez et al. (2022)[16] | Entorhinal cortex | Control, AD | n/a | RIPA buffer lysis. Trypsin digestion | 399 | FC >1.5 and p < 0.05 |
| Bai et al. (2020)[26] | Frontal cortex | Control, pre-clinical AD, MCI, AD | n/a | 8 M urea lysis. Lys-C and trypsin digestion | 20 | FC >1.5 and p < 0.05 |
| Cartyle et al. (2021)[33] | Parietal cortex | Control, pre-clinical AD, AD | Synaptic fraction | 3% SDS lysis. Lys-C and trypsin digestion | 167 | FDR <5% |
| Dai et al. (2018)[17] | Frontal cortex | Control, AD | n/a | 8 M urea lysis. Lys-C and trypsin digestion | 100 | FC >1.5 and p < 0.05 |
| Donovan et al. (2012)[30] | Frontal cortex | Control, AD | Membrane fraction | 8 M urea lysis of insoluble membrane-enriched fraction. Lys-C and trypsin digestion | 9 | FC >1.5 and p < 0.05 |
| Hales et al. (2016)[31] | Frontal cortex | Control, pre-clinical AD, MCI, AD | Insoluble fraction | 8 M urea lysis of sarkosyl-insoluble fraction. In-gel trypsin digestion | 308 | FC >1.5 and p < 0.05 |
| Haytural et al. (2020)[25] | Hippocampus | Control, AD | Specific region: Molecular layer of the dentate gyrus | Laser capture microdissection of specific region. 4% SDS lysis. Lys-C and trypsin digestion | 65 | FDR <5% |
| Higginbotham et al. (2020)[4] | Frontal cortex | Control, AD | n/a | 8 M urea lysis. Lys-C and trypsin digestion | 102 | FC >1.5 and p < 0.05 |
| Ho Kim et al. (2015)[12] | Hippocampus | Control, AD | Specific region: CA4 and dentate gyrus | Manual dissection of specific region. Unreported lysis. Trypsin digestion. | 103 | FDR <5% |
| Hondius et al. (2016)[11] | Hippocampus | Control, AD | Specific region: CA1 and subiculum | Laser capture microdissection of specific region. M-PER lysis buffer plus SDS. In-gel trypsin digestion | 264 | FDR <5% |
| Johnson et al. (2018)[15] | Frontal cortex | Control, pre-clinical AD, AD | n/a | 8 M urea lysis. Lys-C digestion | 841 | ANOVA and Tukey's post hoc p < 0.05 |
| Johnson et al. (2020)[14] | Frontal cortex | Control, pre-clinical AD, AD | n/a | 8 M urea lysis. Lys-C and trypsin digestion | 1081 | ANOVA and Tukey's post hoc p < 0.05 |
| Johnson et al. (2022)[2] | Frontal cortex | Control, pre-clinical AD, AD | n/a | 8 M urea lysis. Lys-C and trypsin digestion | 3493 | ANOVA and Holm post hoc p < 0.05 |
| Li et al. (2021)[21] | Occipital cortex | Control, pre-clinical AD, AD | Soluble fraction; dispersible fraction; SDS- fraction; formic acid fraction. | Fractionation into soluble, dispersible, SDS-soluble and formic acid soluble fractions. Trypsin digestion. | 30 | Kruskal–Wallis H-test p < 0.05 |
| Manavalan et al. (2013)[10] | Hippocampus, Cerebellum | Control, AD | n/a | 2% SDS lysis. In-gel trypsin digestion | 12 | FC >1.5 and p < 0.05 |
| McKetney et al. (2019)[19] | Entorhinal cortex | Control, AD | n/a | 6 M guanidine lysis. Trypsin digestion. | 130 | FC >1.5 and p < 0.05 |
| Mendonca et al. (2019)[28] | Entorhinal cortex, Parahippocampal cortex, Frontal cortex, Temporal cortex | Control, MCI, AD | n/a | 7 M urea lysis. Trypsin digestion. | 938 | FC >1.5 and p < 0.05 |
| Muraoka et al. (2020)[32] | Frontal cortex | Control, AD | Extracellular vesicle fraction | Fractionation into extracellular vesicles. Sonication lysis. Trypsin digestion | 95 | FC >1.5 and p < 0.05 |
| Musunuri et al. (2014)[9] | Temporal cortex | Control, AD | n/a | 1% β-octyl glucopyranoside lysis. Trypsin digestion | 28 | FC >1.5 and p < 0.05 |
| Ojo et al. (2021a)[35] | Frontal cortex | Control, AD | Blood vessel-enriched fraction | Dextran-based cerebrovascular enrichment. 2% lithium dodecyl sulphate lysis. Trypsin digestion. | 33 | FC >1.5 and p < 0.05 |
| Ojo et al. (2021b)[34] | Frontal cortex | Control, AD | Blood vessel-enriched fraction | Dextran-based cerebrovascular enrichment. 2% lithium dodecyl sulphate lysis. Trypsin digestion. | 118 | FC >1.5 and p < 0.05 |
| Pearson et al. (2020)[18] | Ventricle | Control, AD | Membrane and cytosolic fraction | Centrifugation based fractionation. Non-detergent lysis. Trypsin digestion. | 25 | FDR <5% |

**Table 1 (continued) | Studies included in NeuroPro**

| Reference | Brain region(s) | AD stage(s) | Tissue fraction | Lysis and digestion | DEPs included in NeuroPro | Statistical approach |
|---|---|---|---|---|---|---|
| Ping et al. (2020)[23] | Frontal cortex | Control, pre-clinical AD, AD | n/a | 8 M urea lysis. Lys-C and trypsin digestion | 103 | FC >1.5 and p < 0.05 |
| Seyfried et al. (2017)[3] | Frontal cortex, Precuneus | Control, pre-clinical AD, AD | n/a | 8 M urea lysis. Lys-C and trypsin digestion | 937 | ANOVA and Tukey's post hoc p < 0.05 |
| Stepler et al. (2020)[24] | Hippocampus, Parietal cortex | Control, AD | n/a | 8 M urea lysis. Lys-C and trypsin digestion | 81 | FC >1.5 and p < 0.05 |
| Sweet et al. (2016)[13] | Entorhinal cortex | Control, AD | n/a | 2% SDS lysis. In-gel trypsin digestion | 103 | FDR <5% |
| Wang et al. (2020b)[80] | Frontal cortex | Control, AD | n/a | 8 M urea lysis. Lys-C and trypsin digestion | 150 | FDR <5% |
| Wingo et al. (2020)[20] | Frontal cortex | Control, AD | n/a | 8 M urea lysis. Lys-C and trypsin digestion | 856 | FDR <5% |
| Xu et al. (2019)[27] | Hippocampus, Entorhinal cortex, Cingulate gyrus, Motor cortex, Sensory cortex, Cerebellum | Control, AD | n/a | 0.1% SDS lysis. Trypsin digestion. | 2058 | FDR <5% |
| Zellner et al. (2022)[42] | Parietal cortex | Control, AD | Blood vessel-enriched fraction | Ficoll-based extraction of blood vessels. 4% SDS lysis. Trypsin digestion. | 379 | FC >1.5 and p < 0.05 |
| Zhang et al. (2018)[29] | Frontal cortex | Control, AD | n/a | 4% SDS lysis. Trypsin digestion. | 269 | FDR <5% |
| **Amyloid plaques** | | | | | | |
| Drummond et al. (2017)[38] | Hippocampus | AD | Plaques | Laser capture microdissection of amyloid plaques visualised using a combination of 4G8 and 6E10 antibodies. Formic acid lysis. Trypsin digestion. | 1459 | In plaques in >3 cases |
| Drummond et al. (2022)[36] | Hippocampus | AD | Plaques | Laser capture microdissection of amyloid plaques visualised using a combination of 4G8 and 6E10 antibodies. Formic acid lysis. Trypsin digestion. | 2114 | FC >1.5 in plaques vs non-plaques |
| Liao et al. (2004)[39] | Frontal cortex | AD | Plaques | Laser capture microdissection of amyloid plaques visualised using Thioflavin-S. 2% SDS lysis. In-gel trypsin digestion. | 22 | FC >1.5 in plaques vs non-plaques |
| Xiong et al. (2019)[40] | Hippocampus | AD | Plaques | Laser capture microdissection of amyloid plaques visualised using Amylo-Glo. 8 M urea lysis. Trypsin digestion. | 139 | FC >1.5 in plaques vs non-plaques |
| **Neurofibrillary tangles** | | | | | | |
| Drummond et al. (2020)[37] | Hippocampus | AD | Tangles | Laser capture microdissection of tangles visualised using AT8 antibody. Formic acid lysis. Trypsin digestion. | 542 | In NFTs in >3 cases |
| Hondius et al. (2021)[41] | Hippocampus | AD | Tangles | Laser capture microdissection of tangles visualised using AT8 antibody. M-PER lysis buffer plus SDS. In-gel trypsin digestion. | 112 | FC >1.5 in NFTs vs control neurons |
| **Cerebral amyloid angiopathy** | | | | | | |
| Zellner et al. (2022)[42] | Parietal cortex | Control, AD | CAA-containing blood vessels | Ficoll-based extraction of CAA-containing blood vessels. 4% SDS lysis. Trypsin digestion. | 246 | P < 0.05 and FC >1.5 AD vs control |

AD stages refer to cognitively normal controls ('controls'); preclinical AD, mild cognitive impairment ('MCI') and advanced AD ('AD'). Differentially Expressed Proteins ('DEPs') shows the total number of all DEPs included in NeuroPro from each study (sum of DEPs from all AD stages and all brain regions). The statistical approach shows the stringency criteria that was used to define DEPs for each study. All t-tests were two-sided.

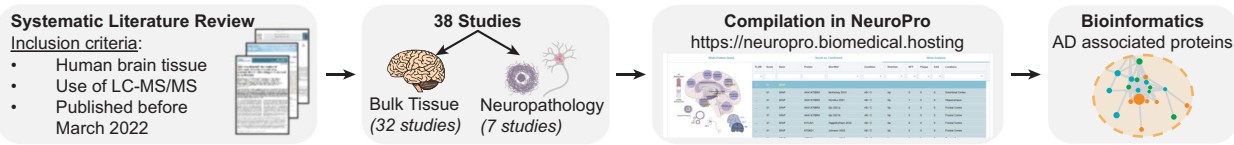

**Fig. 1 | Schematic of methods used in this study.** Note that one study[42] reported two datasets, one of bulk tissue and one of a neuropathological lesion, therefore accounting for the discrepancy between the number of individual studies and the total number of studies.

identified only the most extreme protein differences and missed more subtle protein changes. Importantly, however, lower power studies did not appear to have a higher prevalence of false positive results, only false negative results, meaning that they were still valuable to include in this combined analysis. The high consistency between studies provided us with increased confidence that NeuroPro could be used to examine fundamental unanswered questions about Alzheimer's disease pathogenesis. Key examples of how NeuroPro can be used to examine AD pathogenesis are included in the sections below.

### Neuropathology associated proteins
A unique aspect of NeuroPro is that it includes proteomic data examining the three neuropathological hallmarks of AD: amyloid plaques, NFTs and CAA. This allows a direct comparison of proteins enriched in each neuropathological lesion and provides insight into how protein enrichment in neuropathology and more widespread alteration in bulk tissue are related. Studies included in NeuroPro to date identify 2324 plaque proteins, 615 NFT proteins, and 246 CAA proteins (Supplementary Data 1). Proteins were classified as enriched or depleted in a neuropathological lesion based on a >1.5-fold change difference when compared to surrounding control tissue. Alternatively, proteins were classified as present if they were present in neuropathological lesions in at least three cases within a study.

We were most interested to compare proteins that were enriched/depleted in neuropathological lesions, as these proteins are more likely to be pathologically relevant. We identified 300 proteins enriched in plaques, 125 proteins depleted in plaques, 54 proteins enriched in NFTs, 58 proteins depleted in NFTs, 192 proteins increased in CAA-containing blood vessels and 54 proteins depleted in CAA-containing blood vessels (Fig. 4 and Supplementary Data 1). Only a small subgroup of proteins were commonly enriched in multiple neuropathological lesions (Fig. 4A), suggesting that protein enrichment in neuropathological lesions is selective and not simply a result of the stickiness of Aβ and tau. Only three proteins were commonly enriched in all neuropathological lesions: C4A, CLU and GFAP. As expected, plaques and CAA showed the greatest number of commonly enriched proteins (37 proteins), including many known amyloid-interacting proteins such as APOE, CLU, C3 and C4A. One notable exception was APP: APP was not reported as enriched in CAA in the one CAA-specific study included in NeuroPro, contrasting with the large body of evidence confirming that Aβ is the primary component of CAA. This highlights the important caveat that work in this area is still advancing and proteomic results can be significantly influenced by technical factors (e.g. solubilisation failure in sample preparation or search parameters). Eleven proteins were commonly enriched in plaques and NFTs, likely reflecting the abundant proteins present in phosphorylated tau-rich dystrophic neurites present in neuritic plaques. GO enrichment analysis showed that amyloid plaques were significantly enriched in the extracellular matrix and lysosomal proteins (Fig. 4C and Supplementary Data 4), while NFTs were particularly enriched in neuronal and lumen proteins (Fig. 4D and Supplementary Data 4). There was almost a complete separation of proteins depleted in neuropathological lesions: only one protein—TARDBP (or TDP43)—was commonly depleted in plaques and NFTs (Fig. 4B). The fact that all other depleted proteins were unique to each neuropathological lesion suggests that depleted proteins in

neuropathological lesions are likely cell environment specific responses to each unique type of neuropathology.

A comparison of neuropathology-enriched proteins and bulk tissue studies showed that neuropathology-enriched proteins are not simply those that are also highly enriched in bulk tissue. 214/300 (71%) plaque-enriched proteins, 25/54 (46%) NFT-enriched proteins and 99/192 (52%) CAA-enriched proteins were not consistently altered in ≥5 bulk tissue studies (Fig. 4E). Intriguingly, there were also a small number of proteins that were consistently depleted in AD in ≥5 bulk tissue studies, while being enriched in neuropathological lesions. This shows that bulk tissue studies cannot be directly used to infer neuropathology-specific changes in AD; while there are some consistencies, this is not always the case.

### Protein changes at different clinical stages of AD
We were next interested in identifying high-confidence protein changes that occur in the early clinical stages of AD. This is because these protein changes are likely to be initiating drivers of disease and are potentially drug targets for early AD. Fifteen studies of early AD reached our inclusion criteria for NeuroPro. This included six studies of MCI[26,28,31] and nine studies of preclinical AD[2,3,14,15,21,23,31,33] (Fig. 5A). The proteomic data examining protein changes in preclinical AD is particularly robust, having been obtained from multiple high-powered studies. In contrast, the proteomic data currently available for MCI is less comprehensive and was obtained from lower-powered studies. Based on this limitation, we defined early AD as either preclinical AD or MCI in our analysis below of early-stage AD proteomic changes.

A comparison of the protein changes in early-stage and advanced AD identified 258 protein changes that occurred in both early-stage AD and in ≥5 advanced AD studies. Of these, 240/258 protein changes (93%) occurred in the same direction in both early-stage and advanced AD (Fig. 5A and Supplementary Data 5), suggesting that it is not common for proteins to be increased in early-stage AD and decreased in advanced AD or vice versa. We propose that the 240 protein changes that are consistently altered in the same direction in both early-stage and advanced AD are high-confidence early AD protein changes. Of these early-stage AD protein changes, 99 proteins were increased, and 141 proteins were decreased in AD vs controls. This subgroup of early AD proteins was significantly interconnected ($p < 1.0 \times 10^{-16}$; Protein–protein interaction enrichment). Pathway analysis particularly highlighted early increases in collagen-containing extracellular matrix proteins (Supplementary Data 6). Proteins decreased in early-stage AD were predominantly synapse proteins, which broadly clustered into three groups; those associated with the clathrin vesicle coat (most notably strong enrichment of subunits of the AP-2 adaptor complex), those associated with synaptic vesicles, and those involved in actin filament organisation (Supplementary Data 6). Together, this broadly suggests that there is an early synapse dysfunction in AD, predominantly in glutamatergic synapses.

Pathway analysis also showed that most of the earliest affected cellular components (e.g. synapses, cytoskeleton, lysosome and clathrin vesicles) start with changes in a core cluster of proteins in early AD, which then causes a wave of further protein changes in associated proteins as AD progresses (Fig. 5B). For example, there was strong evidence for exacerbated synapse dysfunction in

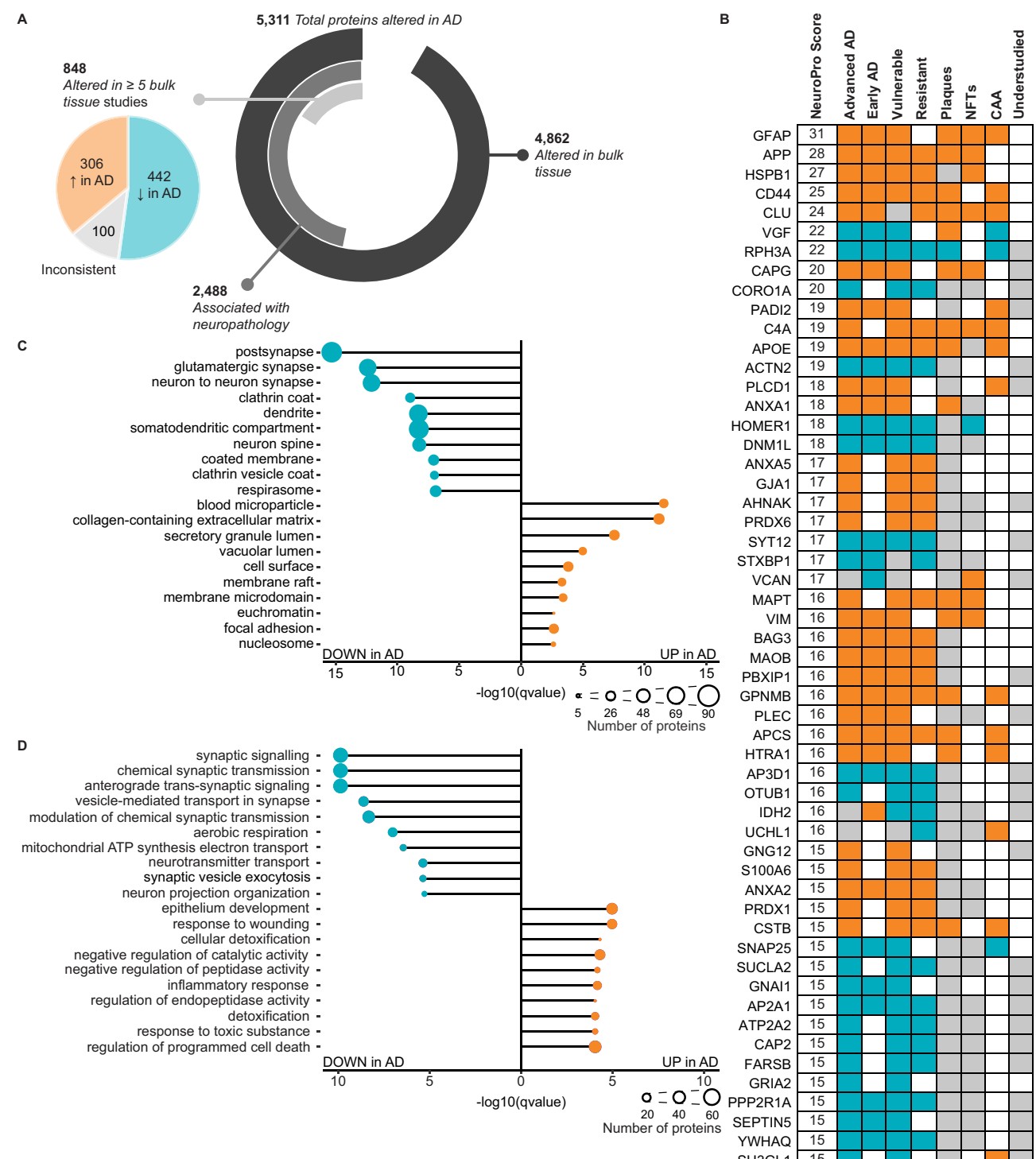

**Fig. 2 | Most consistent protein differences in AD human brain tissue.**
**A** Breakdown of the number of protein differences in NeuroPro. The pie chart shows the breakdown of 848 proteins altered in ≥5 bulk tissue studies into consistently increased, consistently decreased and inconsistent subgroups. **B** 54 most common protein changes in AD. NeuroPro score: number of studies, a protein was significantly altered in. Orange: consistently increased. Blue: consistently decreased. Grey: inconsistently altered. Grey (understudied column): ≤10 studies linking protein with AD. **C**, **D** Most enriched GO terms for proteins (**C** cellular component; **D** biological process) that are consistently upregulated (306 proteins) or consistently downregulated (442 proteins) regulated in AD altered in ≥5 bulk tissue studies. AD Alzheimer's disease, NFTs neurofibrillary tangles, CAA cerebral amyloid angiopathy.

advanced AD, which started with a core cluster of protein changes in early AD. In particular, there was strong evidence for decreased levels of one type of glutamate receptor—AMPA receptors (evidenced by consistent decreases in three of the four protein subunits GRIA1, GRIA2 and GRIA3)—in advanced AD, but not in early-stage AD. Subunits belonging to other glutamate receptors (Kainate and NMDA receptors) were comparatively less affected in advanced AD (Supplementary Data 5). One notable exception to this typical pattern of protein changes through AD progression were widespread changes in mitochondrial proteins, which appeared to be unique to advanced AD (Fig. 5B). In advanced AD, while decreases in proteins from all five complexes of the electron transport chain were

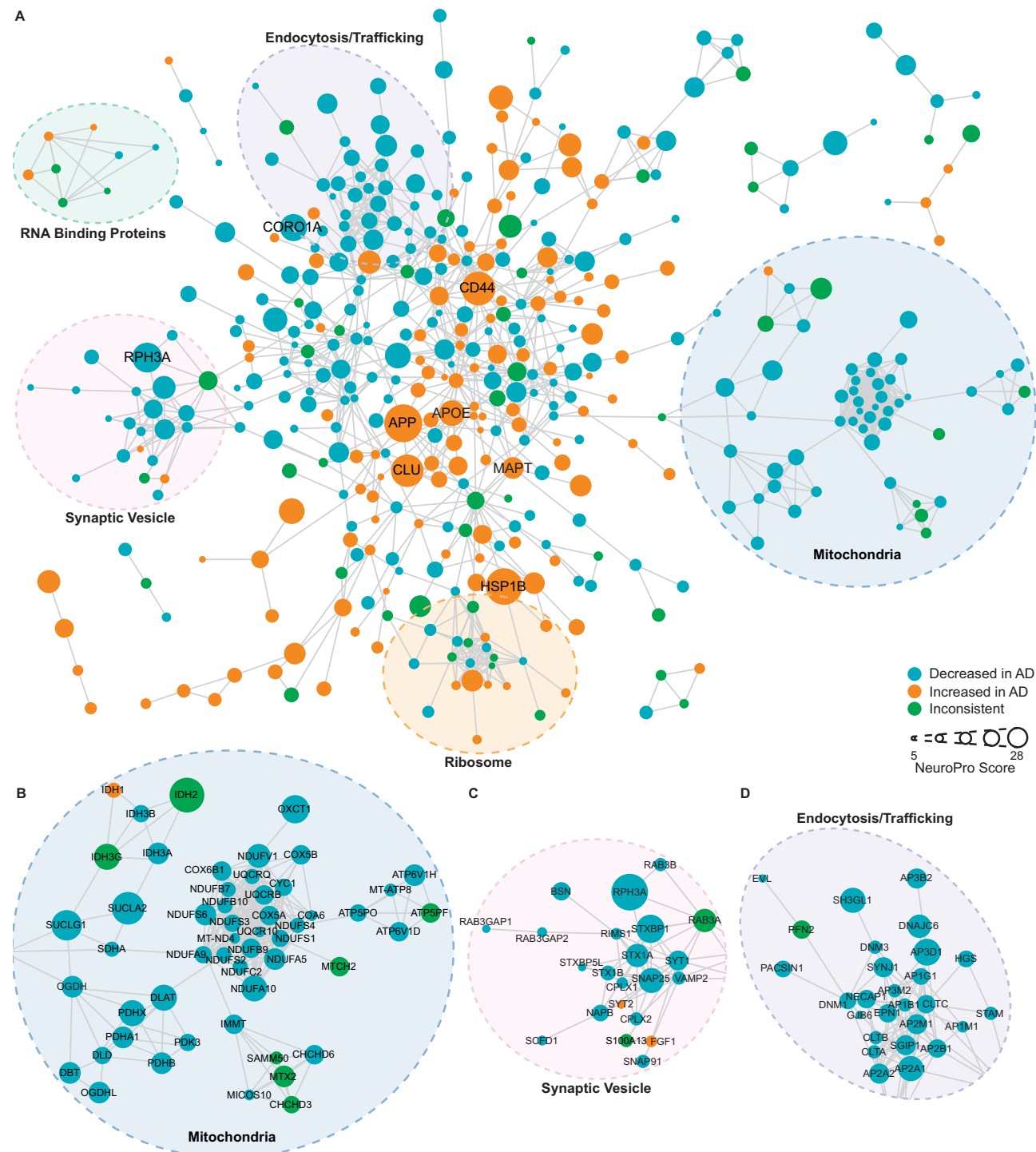

**Fig. 3 | Protein−protein interaction network for 848 proteins altered in ≥5 bulk tissue studies. A** Protein-protein interaction network for proteins altered in ≥5 bulk tissue studies. For simplicity, the network shows high confidence (score >0.7), physical interactions only, and only clusters ≥3 proteins shown. Node size reflects NeuroPro score and colour reflects a consistent directional change in AD (orange: consistently increased; blue: consistently decreased; green: inconsistently altered). Labelled proteins have a NeuroPro score ≥20 or key AD-associated proteins (APOE, MAPT). Higher magnification of three selected sub-networks is shown in **B**−**D**.

observed, this was most prevalent for protein subunits of complex I. Proteins associated with the tricarboxylic acid cycle (TCA cycle), were also selectively decreased in advanced AD, but not early-stage AD (Supplementary Data 6).

Very few proteins were changed in opposite directions in early-stage and advanced AD. Eleven proteins were decreased in early-stage AD but increased in advanced AD and seven proteins were increased in early-stage AD but decreased in advanced AD (Fig. 5A and

Supplementary Data 5). These proteins are particularly interesting as they could represent initially protective protein changes that fail in later disease stages. Remarkably, 5/7 proteins that were increased in early-stage AD and decreased in advanced AD were mitochondrial proteins (PDHB, DBT, NDUFV1, IDH3G and MMUT), supporting a potential influential mitochondrial role in early-stage AD that is driven by select proteins and not widespread mitochondrial protein changes. The 11 proteins that decreased in early-stage AD but increased in

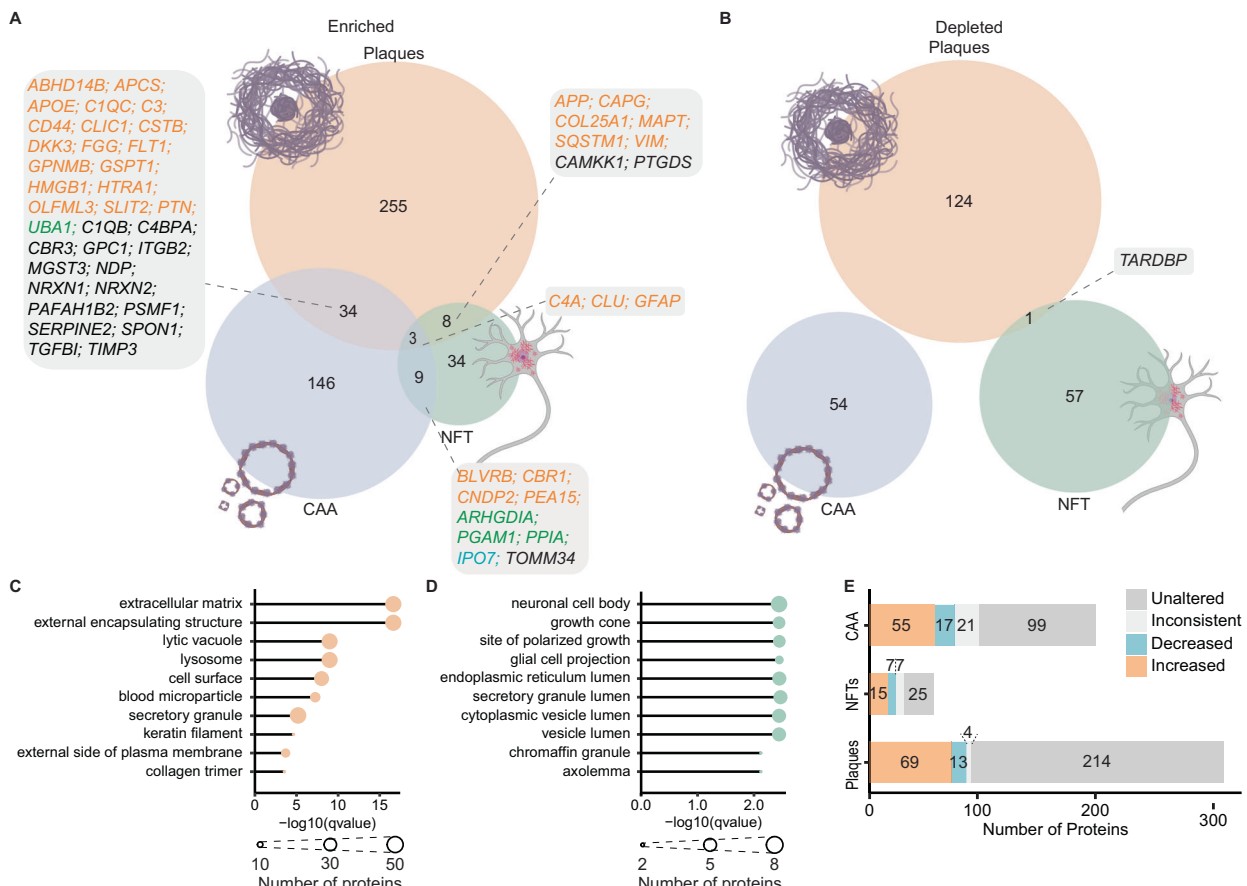

**Fig. 4 | Proteins present in neuropathological lesions. A**, **B** Comparison of proteins enriched and depleted in neuropathological lesions in AD respectively. Boxes specify proteins enriched in multiple lesions; text colour indicates whether proteins are consistently increased (orange), consistently decreased (blue), inconsistently altered (green) in ≥5 bulk tissue studies. Black text indicates proteins either unaltered or altered in <5 studies of AD bulk tissue. **C**, **D** Top enriched GO terms (cellular component) for enriched proteins in amyloid plaques (**C**) and neurofibrillary tangles (**D**). Node size shows the number of lesion-enriched proteins associated with a GO term. **E** Breakdown of directional changes of lesion-enriched proteins in AD bulk tissue. Proteins were increased/decreased if they were consistently altered in ≥5 bulk tissue studies. Proteins were inconsistent if they were inconsistently altered in ≥5 bulk tissue studies. Proteins were unaltered if they were altered in <5 studies of AD bulk tissue. NFTs neurofibrillary tangles, CAA cerebral amyloid angiopathy.

advanced AD showed no enrichment of any pathway, function or cellular component.

## Region-specific protein changes in Alzheimer's disease

Next, we were interested in identifying proteins that were linked to selective vulnerability in AD. For this analysis, we used NeuroPro to compare protein changes in advanced AD in 12 brain regions, which were classified as vulnerable or resistant brain regions in AD (Fig. 6A and Supplementary Data 7). We were particularly interested in determining if there were consistent protein changes in brain regions that are vulnerable and resistant to AD. After filtering for high-confidence protein changes, 510 proteins were identified as altered in vulnerable brain regions in advanced AD (Fig. 6A). 40% (203/510) of these proteins were altered in the same direction in both resistant and vulnerable brain regions (90 proteins were increased in both; 103 proteins decreased in both; Fig. 6B), providing evidence of AD-associated protein dysfunction in resistant brain regions that do not have widespread neuropathology.

21% (64/307) of protein changes that were uniquely present in vulnerable brain regions, but not resistant brain regions were proteins enriched in neuropathological lesions (e.g. GFAP, APP, VGF, HTRA1, MDK, SMOC1 and SQSTM1), confirming that neuropathology is not a predominant feature in resistant regions in advanced AD (Supplementary Data 7). Very few proteins showed opposite protein changes in vulnerable regions vs resistant

regions (ten proteins; all decreased in vulnerable regions but increased in resistant regions).

Based on the significant overlap in protein changes in vulnerable and resistant brain regions in advanced AD, we hypothesised that protein changes in resistant brain regions in advanced AD (e.g. sensory cortex, motor cortex and cerebellum) may be the same as those in vulnerable regions in early-stage AD. If so, this would reflect a temporal wave of progressive protein dysfunction through affected brain regions as AD progresses. To test this hypothesis, we directly compared protein changes in resistant regions in advanced AD with protein changes in vulnerable regions in early-stage AD (Supplementary Data 8). This analysis showed considerable overlap between these two groups of proteins, supporting our hypothesis. Sixty-four proteins were altered in the same direction in the two datasets (Fig. 7A and Supplementary Data 8); 35 proteins were decreased in AD and 29 proteins were increased in AD. We propose that these protein changes are some of the earliest protein changes in AD, occurring prior to the development of neuropathology and persisting throughout disease progression. Notably, while this subset of pre-neuropathology protein changes includes many well-known AD-associated proteins (e.g. APOE, MAOB and AQP4), it does not include APP and MAPT. This reflects the fact that widespread neuropathology is not yet present in resistant brain regions. Pathway analysis of pre-neuropathology protein changes highlighted increased levels of chaperones associated with aggregated Aβ and tau (HSPB1, HSPB8, BAG3, APOE and APCS),

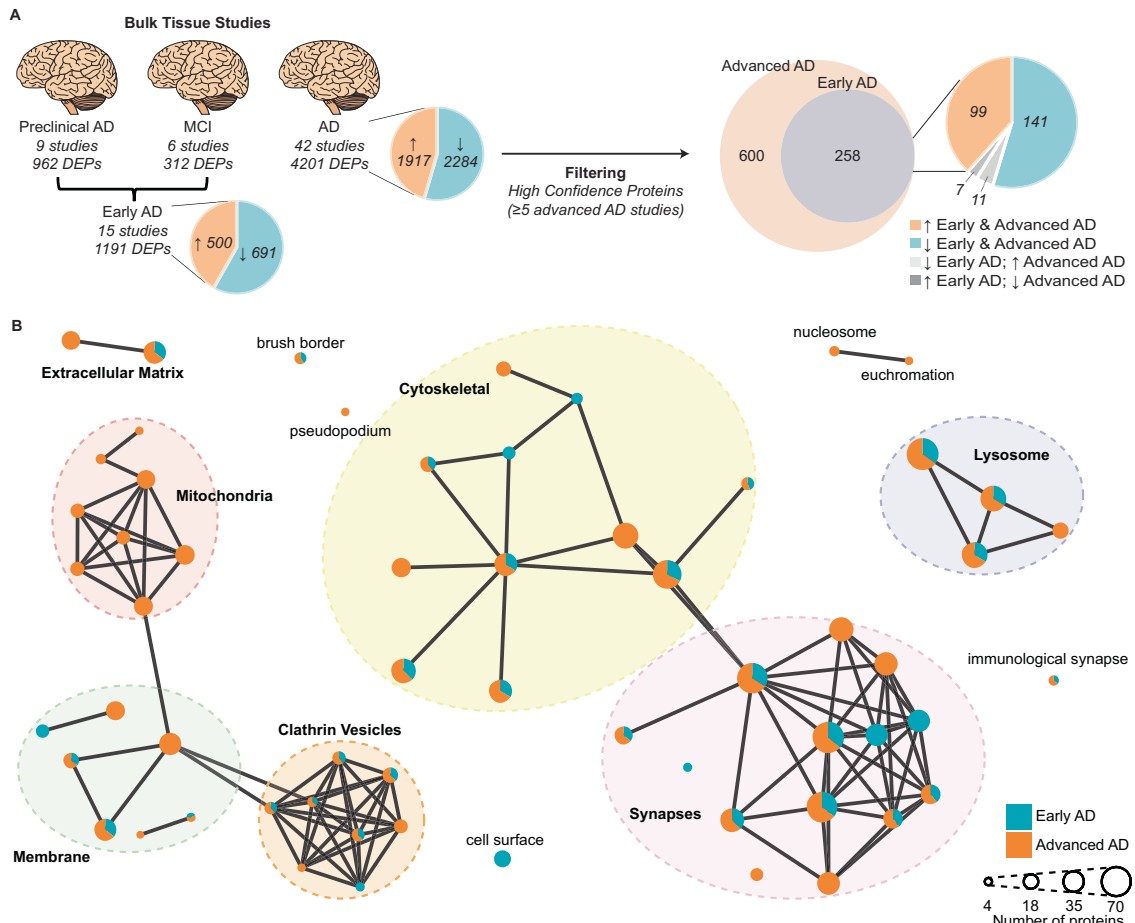

**Fig. 5 | Comparison of protein changes in early-stage and advanced AD.**
**A** Filtering method used to identify high-confidence early-stage and advanced AD protein changes. Proteins that were consistently altered in the same direction in each of the three AD clinical stages were identified (DEPs). Inconsistent proteins within a clinical stage were removed. Preclinical AD and MCI were combined to generate an early AD group. Pie charts specify the number of proteins that are increased/decreased in AD vs controls. Protein changes were filtered to identify high-confidence protein changes in early-stage and advanced AD (altered in ≥5 bulk tissue studies). **B** GO term (cellular component) network clusters, mapped to show clusters of similar GO terms which were manually annotated to show the overarching cellular component linked to major clusters of GO terms. Each node represents a unique GO term. Node colour shows the proportion of proteins for each enriched GO term that were altered in early-stage AD (blue) and advanced AD (orange). AD Alzheimer's disease, MCI mild cognitive impairment, DEPs differently expressed proteins.

increased levels of enzymes associated with energy production and biosynthesis of neurotransmitters (SPR, PKM, PAICS, MTAP, MAOB, LTA4H, GYG1, BBOX1 and ALAD) and proteins involved in innate immunity. A significant subgroup of these pre-neuropathology protein changes were synaptic proteins (24/64 proteins; Fig. 7B). These altered synaptic proteins were associated with many intracellular components in both the pre- and post-synapse (Fig. 7B), suggesting widespread early synaptic dysfunction in AD. Based on this analysis we propose that there are three phases of protein changes in AD human brain tissue: Phase 1 (pre-neuropathology protein changes), Phase 2 (Early-stage AD protein changes which occur early in the disease alongside neuropathology development) and Phase 3 (Advanced AD protein changes) (Supplementary Data 9).

## Discussion

NeuroPro provides a comprehensive roadmap of protein changes that occur in the human brain throughout the progression of AD. NeuroPro provides insight into AD pathogenesis and highlights potential drug targets and biomarkers for AD. As such, we propose that it is a useful innovative resource for the AD field. Its novelty lies in our combined analysis approach of diverse mass spectrometry datasets that often have limited power when analysed in isolation. Our online database allows users to immediately place AD brain protein changes in the

context of clinical disease stage, brain region specificity, association with neuropathology and subcellular/biochemical changes in disease (e.g. subcellular localisation or insolubility in disease). To demonstrate the power of NeuroPro, we have used it here to examine key questions about AD pathogenesis. In doing so, we have: (1) shown that proteomic studies of AD tissue are highly consistent, (2) shown that proteins enriched in plaques, NFTs and CAA are largely unique and independent of broader protein changes in bulk tissue, (3) shown that there are many similar protein changes in resistant brain regions in AD and early clinical stages of AD (4) identified some of the earliest protein changes in AD, including a subset that we hypothesise to occur prior to neuropathology development. These results are just a few examples of the type of analyses that can be performed using NeuroPro. An additional key benefit of NeuroPro is that users can search their own datasets. NeuroPro provides immediate context to newly generated data about the involvement of protein hits in AD and allows rapid comparison between protein changes in AD and other diseases.

One of our key findings was that widespread decreases in synapse proteins appear to be one of the earliest pathological changes observed in human AD brain tissue. While synaptic changes are known to have a pivotal role in AD[45,46] and have been reported in both pre-clinical AD and MCI[47–49], exactly how early synaptic dysfunction occurs in human AD and the initiating protein drivers involved are unknown.

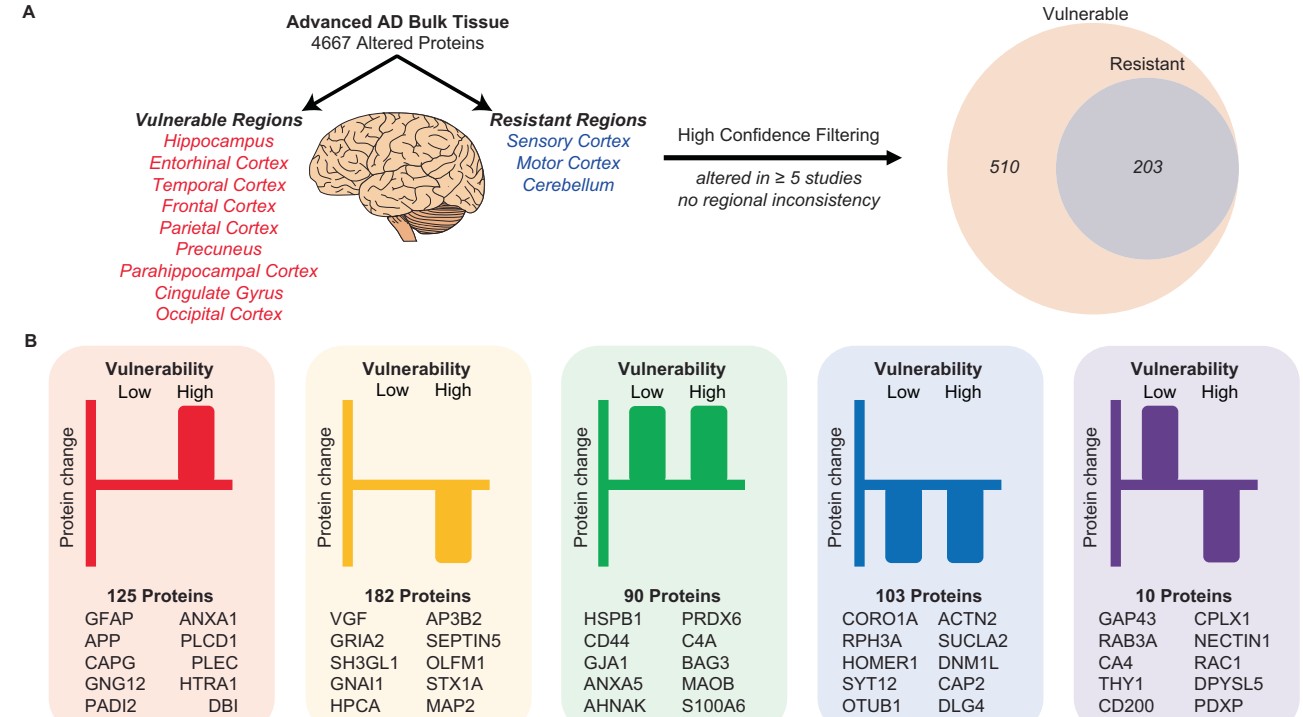

**Fig. 6 | Comparison of protein changes in vulnerable and resistant brain regions. A** Filtering methodology used to identify high-confidence protein changes observed in vulnerable and resistant brain regions. Proteins were only considered high-confidence protein changes if they were altered in ≥5 bulk tissue studies. **B** distribution of protein changes into five types of protein changes: proteins increased (red) or decreased (yellow) in vulnerable regions only, proteins increased (green) or decreased (blue) in both vulnerable and resistant regions, proteins increased in resistant regions and decreased in vulnerable regions (purple). Proteins with the highest total NeuroPro score in each group are specified. AD: Alzheimer's disease.

We identified a subgroup of 24 synapse proteins that were significantly altered in resistant brain regions prior to the development of local AD neuropathology, suggesting that these synaptic protein changes could be among the first pathological changes in AD. We hypothesise that these initial synaptic protein changes promote downstream synaptic dysfunction in neighbouring proteins, interactors or members of the same networks as AD progresses, which is reflected in the increased number of synaptic protein changes at later stages of AD. These pre-neuropathology synapse protein changes were observed throughout both the pre- and post-synapse and were associated with many sub-cellular compartments, suggesting widespread dysfunction throughout the synapse in early AD. While little is known about how most of these proteins are mechanistically involved in AD, the role of some of these proteins in AD has been previously explored: for example, DNM1L[50,51], SYNPO[52] and YWHAZ[53,54] all appear to promote AD-associated pathology. Mechanistic roles for other non-synaptic protein changes in this pre-neuropathology phase have also been reported, strengthening support for the potential importance of this subset of proteins. For example, there is evidence that HSPB1[55], BAG3[56], GPNMB[57], AQP4[58], HSPB8[59,60], PKM[61], DKK3[62], GLRX[63] and GAS7[64] are protective in AD, while MAOB[65] and CD44[66] appear to promote AD-associated pathology. These previous studies confirm that many of these pre-neuropathology protein changes are potential drivers of AD and suggest that the mechanistic roles of the remaining proteins in this group should be examined in future studies.

Another key finding was that enrichment of proteins in neuro-pathological lesions is a selective process that is unique to each specific type of lesion. We hypothesise that the enriched proteins present in each type of lesion reflect the processes involved in lesion development. For example, plaques were significantly enriched in lysosomal proteins, nicely complementing recent studies that suggest that amyloid plaques form after the accumulation of intraneuronal Aβ in autophagic vacuoles[67]. In contrast, NFTs were significantly enriched in neuronal and endoplasmic reticulum proteins, supporting previous studies showing a strong association between tau and ribosomal proteins that can pathologically impair translation[68,69]. In addition, our results highlight the importance of localised proteomics approaches[70–72] to identify neuropathology-enriched proteins, as protein enrichment in neuropathology was often not reflected in bulk tissue studies.

We were intrigued to see that many AD-associated protein changes were consistently observed in the same direction in early AD, advanced AD, vulnerable brain regions and resistant brain regions. Together, our results suggest that resistant brain regions in advanced AD may develop the same protein changes observed in vulnerable brain regions in early AD; supporting a hypothesis that there may be a temporal wave of progressive protein changes throughout the brain in AD. While it is well established Aβ and tau progressively spread through the brain in AD[73,74], here we show that this same process may occur for many other proteins also. Importantly, a subset of these protein changes (many of them synaptic proteins) appears to occur before Aβ and tau accumulation, suggesting that they may be early disease drivers. Our results support the concept that resistant brain regions are initially protected against pathology (possibly due to a range of factors[75–77]); however, once these protective measures eventually fail, the same initiating protein changes observed in vulnerable brain regions in early AD become apparent. If true, this has potential implications for experimental studies as resistant brain regions in advanced AD could be used as a proxy to study early AD-associated protein changes in human brain tissue. Given these implications, these results should be further explored in future studies to confirm our preliminary findings proposed here.

There are several limitations to our study, which highlight critical future research directions. For example, only a limited number of MCI

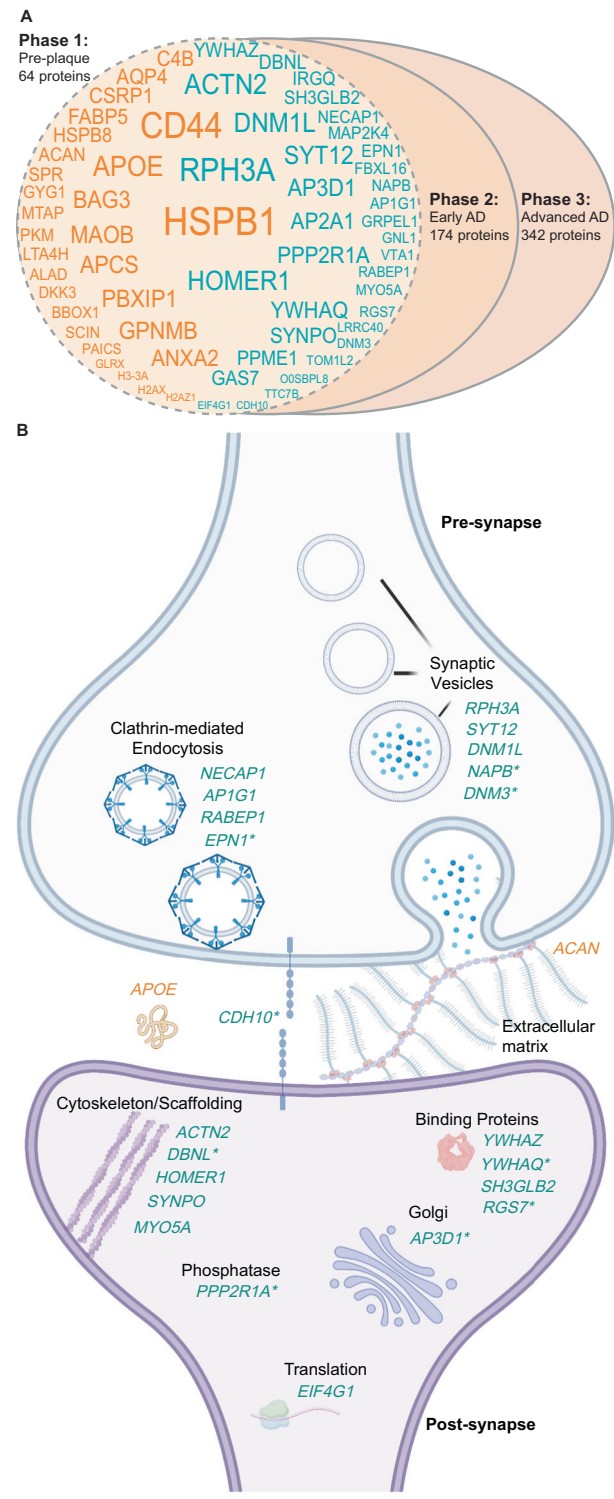

**Fig. 7 | Proposed pre-neuropathology protein changes. A** High-confidence protein changes were mapped to three successive phases of the disease: pre-neuropathology phase, early AD phase and advanced AD phase. The 64 proposed pre-neuropathology protein changes are identified by gene IDs. Text size reflects the total number of studies each protein has been reported to be significantly altered in (i.e. NeuroPro score). Orange text: proteins consistently increased in AD vs controls; blue text: proteins consistently decreased in AD vs controls. **B** 24 pre-neuropathology protein changes were synaptic proteins. Schematic highlights simplified key locations of each of these 24 altered proteins based on GO terms, noting the caveat that many of these proteins have multiple functions and locations within the synapse, which is not represented here. Twenty-two synaptic proteins were downregulated (blue text), while two synaptic proteins were upregulated (orange text). * indicates proteins that are reported to be located in both the pre- and post-synapse. For simplification, each protein is only highlighted once in either the pre- or post-synapse. AD Alzheimer's disease.

other brain regions. Additional high-powered studies examining other brain regions, particularly resistant brain regions, would be exceptionally useful to determine the temporal progression of protein changes throughout the progression of AD. In addition, more studies are required to confirm proteins enriched in neuropathological lesions. While proteins that are present in neuropathological lesions are interesting, proteins that are selectively enriched in neuropathological lesions in comparison to neighbouring control tissue are more likely pathology drivers. In particular, additional datasets examining CAA are needed given that the current single dataset that met our inclusion criteria did not identify Aβ and other major amyloid-associated proteins[42], which we propose to be due to the lack of solubilisation of Aβ in these samples because of the lack of formic acid treatment during sample preparation prior to mass spectrometry in this study. Looking forward, NeuroPro will continue to expand as proteomic studies increase in the future. There is potential for the inclusion of proteomic studies of disease models (such as iPSCs or animal models), which would be useful for addressing important questions about how closely these models reflect human AD. With regards to iPSCs, it would be important for future studies to compare the proteome of multiple unique patient-derived lines from both AD and controls to reflect the inter-patient variability typically present in human brain tissue studies. Additionally, NeuroPo could be expanded to include changes in protein post-translational modifications (e.g. phosphorylation, acetylation, ubiquitination etc.), which are known to be critical disease mediators in AD. Only a small number of proteomic studies examining post-translational modifications in human AD brain tissue have been performed to date; however, as studies in this area increase, these would be an excellent future addition to NeuroPro.

To conclude, we have shown that NeuroPro is a powerful resource that provides insight into AD pathogenesis and highlights many novel or understudied proteins in the AD field, providing exciting avenues of research for future studies. This has the potential to increase the impact and widespread use of proteomic data and will hopefully pave the way forward for new therapies and biomarkers for AD.

## Methods
### Systematic literature review and inclusion/exclusion criteria for NeuroPro
A comprehensive literature search was performed to identify all LC-MS/MS studies of human AD brain tissue. The following search terms were used on Pubmed: 'Alzheimer's proteomics' and 'mass spectrometry human brain Alzheimer's'. All papers published prior to March 15, 2022, were included in the search. In addition, our own accepted manuscript[36] was also manually included. Search results were automatically filtered on Pubmed prior to manual screening using the English and Human filters and reviews were excluded. Results were then manually screened by one reviewer (ED) to identify acceptable studies that defined protein changes in bulk tissue homogenate

proteomic studies were available, and these were largely underpowered. Future, high-powered proteomic studies comparing protein changes in MCI and preclinical AD tissue are needed, as these could be used to specifically identify protein changes linked to cognitive impairment while controlling for neuropathology. This is additionally important because there is still debate about whether preclinical AD cases are, in fact, early AD, as they could instead be individuals resilient to AD. Future high-powered proteomic studies directly comparing preclinical AD to MCI are needed to address this important question. Furthermore, most AD proteomics studies to date have examined protein changes in the frontal cortex: only a handful have examined

between (i) AD vs controls, (ii) mild cognitive impairment (MCI) vs controls and (iii) preclinical AD (also referred to in the literature as asymptomatic AD, prodromal AD or high pathology control) vs controls. In line with the current literature, we defined preclinical AD as the presence of amyloid plaques in the absence of cognitive impairment[78]. Inclusion criteria for bulk tissue homogenate studies were: analysis of human brain tissue, use of LC-MS/MS, and sufficient accessible LC-MS/MS data. Proteins were considered to be differently expressed between disease and control (DEPs) based on any of the following statistical approaches: FDR of <5%, $p < 0.05$ using ANOVA and an appropriate post hoc test (e.g. Tukey's or Holm's comparison post hoc test), $p < 0.05$ using Kruskal–Walis $H$-test, or a combination of $p < 0.05$ using $t$-test combined with a fold change difference >1.5-fold. Studies using 2D-gel electrophoresis were excluded. Studies that did not specify brain regions (e.g. analysed proteins in 'cortex') were excluded. Studies that did not provide a protein identifier (or an adequate identifier that could be used to map a corresponding protein identifier) in their datasets were excluded.

Data were manually collected from published manuscripts and their supplementary data by a single reviewer (ED). Data collected from each study included lists of all proteins identified in the study and accompanying statistical and fold change data (if available) or similar data for significantly altered proteins if the full list of identified proteins was not provided. Datasets were manually adjusted to permit direct comparison between studies using the following methods: Single gene ID and UniProt ID were generated for each reported protein; if multiple gene IDs for a single protein were provided, the first listed Gene ID was used. UniProt IDs were stripped of isoforms. Duplicate gene IDs within a dataset were removed. p-values (generated using unpaired, two-sided $t$-test) and fold change differences between AD and control groups were manually performed using published data if not provided in the original study and sufficient data were available. Proteins identified by only one peptide were excluded. Published lists of DEPs were manually filtered to only include those that reached our stringency criteria detailed above.

Proteomics studies examining the proteome of amyloid plaques, neurofibrillary tangles (NFTs) or cerebral amyloid angiopathy (CAA) were also included. Inclusion was limited to studies that selectively isolated neuropathological lesions using either laser capture microdissection (for plaques and NFTs) or a dextran/Ficoll gradient extraction (for CAA-containing blood vessels). A protein was considered present in plaques or NFTs if it was present in ≥3 cases within a study. Reported results were manually filtered to exclude proteins identified by only 1 peptide. Proteins were considered increased or decreased in a plaque, NFT, or CAA-containing blood vessel based on >1.5-fold change difference between plaques:non-plaque regions, NFT containing neurons:non-NFT containing neurons or CAA-containing blood vessels:non-CAA-containing blood vessels.

## NeuroPro database

Data from selected studies were uploaded into NeuroPro: https://neuropro.biomedical.hosting. All proteins were annotated with both a protein identifier (UniProt ID) and gene ID. Proteins were grouped within NeuroPro using GeneID. The entire NeuroPro dataset can be downloaded in the meta-analysis tab within the NeuroPro database. A NeuroPro Score for each protein was generated based on the number of times that protein was reported as significantly altered in AD in published studies. In NeuroPro, proteins can be filtered according to brain region, disease stage, association with neuropathology or direction of change in AD, either alone or in combination in the meta-analysis tab. The resulting filtered datasets can be exported. Users can also directly compare datasets of interest (e.g. user-generated data or published proteomic data examining a different disease) in the Multi-Protein Query tab, which categorises searched proteins into known or novel AD-associated proteins in the Novel vs. Confirmed tab.

## Analysis of NeuroPro

**Consistency of directional change.** Data was exported from NeuroPro for analysis on 7-21-22 (Supplementary Data 1). The NeuroPro score for each protein was obtained from NeuroPro and an additional NeuroPro (Bulk Tissue) Score was generated, which was a count of the number of times a protein was designated a DEP in studies of bulk tissue homogenate only (i.e. excluding studies of neuropathological lesions). Proteins were designated as Increased/Decreased in AD if they were consistently altered in the same direction in ≥5 studies of bulk tissue homogenate, with one outlier permitted (to accommodate the range of brain regions, fractions, and disease stages). DEPs in ≥5 studies of bulk tissue homogenate with two or more outlier directional changes were designated Inconsistent in AD.

**Comparison of protein changes in early-stage vs advanced AD.** Bulk tissue data were exported from NeuroPro. DEPs were filtered into those occurring in preclinical AD, MCI and advanced AD and individual NeuroPro Scores were assigned for each clinical stage of AD. Proteins were designated as Increased or Decreased in each clinical stage of AD if they were consistently altered in the same direction in all studies within that clinical stage of AD. One-directional outlier was permitted only in cases where a protein was altered in ≥5 studies within a clinical stage. Proteins that were increased in AD received a positive score and proteins that were decreased in AD received a negative score. All other proteins were classified as having Inconsistent directional change within that clinical stage of AD and did not receive a score. The resulting dataset was then filtered to only contain proteins with a score in at least one stage of AD (Supplementary Data 5).

Protein changes were compared in early-stage AD (defined as either preclinical AD or MCI) and advanced AD. Advanced AD protein changes were restricted to include high-confidence protein changes only, which were classified as consistently present in ≥5 studies of advanced AD (one directional outlier permitted). Protein changes were grouped into those that were (1) altered in the same direction in both early-stage and advanced AD (proposed early-AD protein changes), (2) uniquely altered in advanced AD and not early-stage AD (proposed advanced AD protein changes), (3) altered in the opposite direction in early-stage and advanced AD or altered only in early-stage AD and not in advanced AD (proposed opposite protein changes in advanced AD and Early-AD). All other proteins were classified as Unable to group. This dataset was then compared to MitoCoP—a dataset of high-confidence human mitochondrial proteins[79] and SynGo—a dataset of high-confidence human synaptic proteins (2021 download; https://www.syngoportal.org/), to highlight mitochondrial and synaptic proteins, respectively.

**Comparison of protein changes in different brain regions.** Bulk tissue data examining advanced AD was exported from NeuroPro. This analysis was restricted to advanced AD as current proteomic studies of early-stage AD tissue have largely been restricted to the frontal cortex, therefore precluding an in-depth regional comparison. Advanced AD proteomic data was available for 13 brain regions; frontal cortex, hippocampus, parahippocampal cortex, entorhinal cortex, temporal cortex, parietal cortex, cingulate gyrus, precuneus, motor cortex, sensory cortex, occipital cortex, ventricle wall and cerebellum. Brain regions were designated as either vulnerable or resistant in AD based on consensus in the literature about the timing and extent of neuropathology. Vulnerable regions included: the entorhinal cortex, hippocampus, parahippocampal cortex, temporal cortex, frontal cortex, parietal cortex, precuneus, cingulate gyrus and occipital cortex. Resistant regions included: the sensory cortex, motor cortex and cerebellum. Data examining proteomic changes in the ventricle wall[18] were not included in this analysis as there is not sufficient literature available to determine whether this region is vulnerable or resistant to AD. A brain region-

specific NeuroPro score was generated for each region, which was a count of the number of studies where that protein was reported to be significantly altered in AD within that brain region. Proteins consistently increased in AD vs controls received a positive score and proteins consistently decreased in AD vs controls received a negative score.

An analysis of protein differences in Vulnerable and Resistant brain regions was performed (Supplementary Data 7). High-confidence protein differences were identified as those with a combined region score of ≥5. These high confidence protein changes were then categorised as; (1) Increased in Vulnerable; Unchanged in Resistant, (2) Decreased in Vulnerable; Unchanged in Resistant, (3) Increased in Vulnerable and Resistant (4) Decreased in Vulnerable and Resistant, (5) Increased in Vulnerable; Decreased in Resistant, (6) Decreased in Vulnerable; Increased in Resistant and (7) Inconsistent.

**Analysis of the pre-neuropathology AD protein changes.** Pre-neuropathology protein changes were designated as those consistently present in the same direction in resistant brain regions, early-stage AD and advanced AD (Supplementary Data 8, 9). This subset of pre-neuropathology protein changes was identified by direct comparison of early-AD protein changes (Supplementary Data 5) and protein changes in resistant brain regions (Supplementary Data 7). Proteins that were consistently altered in both early-stage AD and advanced AD, but not in resistant brain regions were designated as Early AD protein changes and protein changes that were consistently altered only in advanced AD were designated as Advanced AD protein changes. Protein changes that were altered in an inconsistent direction between any of the protein subsets analysed (resistant protein changes; early AD protein changes; advanced AD protein changes) were classified as inconsistent protein changes and excluded from analysis (Supplementary Data 10).

**Pathway/network analysis or comparison with previous datasets**
General data manipulations and grouping were performed in R v4.0.2 using the tidyverse v1.3.2 collection of packages. Plots were generated in R with the packages ggplot2 v3.4.0, ggpubr v0.5.0, ggrepel v0.9.3, and all figures were edited in Adobe Illustrator v27.1.1. Gene Ontology (GO) enrichment analysis was performed in R using the packages enrichplot v1.16.2, clusterProfiler 4.4.4, using the genome-wide annotation for human, org.Hs.eg.db v3.15.0. Prior to analyses gene IDs were mapped to Entrez IDs with the 'bitr' function of clusterProfiler. GO terms were filtered to an FDR <0.05 and the full list of proteins detected were used as the background list (5,311 proteins). Primarily the GO cellular compartment (GOCC) and GO biological process (GOBP) annotations were used. GO terms were reduced with the 'simplify' function from clusterProfiler to reduce heavily redundant terms prior to plotting. GO terms were plotted as either top ten 'lollypop' plots or enrichment maps where nodes (GO annotations) are connected by shared proteins. Protein–protein interaction networks were generated in STRING v11.5 and the networks were edited in Cytoscape v3.9.1 and Adobe Illustrator. Pathway collections were annotated manually based on string gene ontology outputs. Venn diagrams were generated with the R package Venerable v 3.1.0.9000 and edited in Adobe Illustrator. Selected figure panels were created with BioRender.com.

**Comparison with previous literature**
Systematic Pubmed searches were performed to determine a particular protein's known association with AD. Search terms used were "protein name" or "gene ID" Alzheimer's", and the search was performed on 15-9-22. The protein name was obtained from UniProt.

**Reporting summary**
Further information on research design is available in the Nature Portfolio Reporting Summary linked to this article.

## Data availability
All data used in this study was obtained from prior publications that were obtained from Pubmed (listed in detail with accompanying citations in Table 1). All data used in this study is available on the open-access NeuroPro website: https://neuropro.biomedical.hosting. All data used in sub-analyses are included in full in the supplementary data of this manuscript.

## Code availability
No new code was generated in this manuscript. All software used is open source publicly available or licenced and options used beyond the default are noted in methods.

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

## Acknowledgements

This study was supported by funding from NIH (P01AG060882 and P30AG066512) to T.W., Bluesand Foundation to E.D., Alzheimer's Association (Blas Frangione Early Career Award) to E.D., and the National Health and Medical Research Council of Australia (Programme Grant # 1132524) to T.K.

## Author contributions

E.D. conceived and designed this study and was responsible for the identification of studies. E.D. and M.A. were responsible for data curation and combined analysis. M.A. designed and maintains the NeuroPro database. T.K. and E.D. were responsible for bioinformatics analyses and figure generation. G.P., T.W., and B.U. provided expert advice on the interpretation of data. E.D. wrote the manuscript with input from co-authors. All authors read and approved the final manuscript.

## Competing interests

The authors declare no competing interests.
