## [Peer Review File · Nature Communications]

Compilation of all known protein changes in the human
Alzheimer's disease brainREVIEWERS' COMMENTS

Reviewer #1 (Remarks to the Author):

The manuscript by Askenazi et al. provides a meta-analysis of previous proteomic studies analyzing protein changes in the Alzheimer's disease (AD) by mass spectrometry. The authors compare differentially expressed proteins in preclinical AD, mild cognitive impairment, and advanced AD in different brain regions as well as neuropathologic lesions. Among the main findings are: 1) a high inter-study consistency showing that protein changes in the AD brain occurs in the same direction (up- or downregulation), regardless of brain region, 2) bulk brain tissue and neuropathologies did not show highly similar proteomic changes, indicating that bulk tissue studies cannot be directly used to infer neuropathology changes in AD, 3) identification of protein changes in early vs advanced AD, thus potentially identifying initiating drivers of the disease and drug targets for early AD, 4) almost half of the proteins with the greatest changes in expression in AD are understudied in the AD field with 4 proteins identified that have not been directly linked to AD prior, 5) the earliest affected cellular components were synapses, cytoskeleton, lysosome, and clathrin vesicles, while changes in mitochondrial and neuropathologic lesion proteins were associated with advanced AD, 6) protein changes in vulnerable brain regions in early stages of AD are similar to changes in resistant brain regions in advanced AD, suggesting a temporal wave of progressive dysfunction in different brain regions during AD progression, and 7) the establishment of a searchable database (NeuroPro) that provides a new resource for researchers. Meta-analyses of proteome changes in AD brain have been published before and do, by themselves, not offer a novel concept of analysis. However, when a meta-analysis of other primary as well as meta-analyses studies helps to overcome limitations of smaller study sizes, then there certainly is great value. This manuscript combines 38 published studies with 59 unique comparisons and represents the largest analysis thus far. Askanazi et al. is without a doubt an important study with great significance that lacks major flaws in data analysis, interpretation, and conclusions.

Specific minor points:

- 1) The introduction nicely outlines tissue fractions from which protein differences have been evaluated, "insoluble, synaptic, membrane, or blood vessel-enriched fractions". A column with this information would be useful for the "Sources" table provided to the NeuroPro website
- 2) Preclinical should be better defined to guide the reader and is not recommended to be shortened to an acronym.
- 3) Some proteins such as VGF and SH3GL1 were decreased in AD brain tissue but enriched in neuropathologic lesions. The authors conclude that the "results suggest that VGF and SH3GL1 may have unique roles in AD pathogenesis." Consideration of protocol differences between preparation of bulk tissue and neuropathologic lesions should be discussed, or included as a limitation of interpretation of different protein identifications.
- 4) One limitation is that this study compares differentially-expressed proteins, but does not look at changes in protein modifications (including phosphorylation) which undoubtedly are critical in regulating protein activity, transport etc. Some text in the Discussion regarding this for future studies or text regarding in limitation section.
- 5) The authors address that animal studies are of limited value for incorporation into NeuroPro, but some Discussion of proteomics in iPSCs may be of interest for future studies or as a resource page on NeuroPro.
- 6) Antibody usage to identify neuropathologic lesions should be included in the Sources table on NeuroPro to help guide the user.
- 7) Figure 1, second panel says 38 studies but then breaks down into 32 and 7. Was there an overlapping study that should be commented upon in the figure legend? The date in Figure 1 is also recommended to be updated to March 2022 as the formatting may not be intuitive to all

readers.

8) Why do the authors go from supplementary table 2 to supplementary table 8? An then jumped from Supplementary Table 3 to Supplementary Table 10

9) Figure 4 is a great summary figure!

10) Generally it is recommended text for "late AD" be updated to "advanced AD". With LATE-NC representing TDP-43, the use of "late AD" may cause confusion. This is recommended for text and figures. Figure 5A says "late AD studies" under arrow but elsewhere it says "advanced AD." Supplementary Table 10 says "late AD" – change to "advanced AD." These should be changed for consistency.

11) Figure 6B could be more clear in the labelling where it says vulnerability.

12) Can future proteomic studies be added to the NeuroPro website with either user recommendations or a yearly review of literature?

13) Perhaps update the "Multi-Protein Query" so that it is more user friendly. For example, change it to where the user can click on an experimental protein set to look at novel vs confirmed proteins instead of typing in/deleting protein sets. Or include a reset button to get the experimental protein sets to repopulate after deletion.

Reviewer #2 (Remarks to the Author):

Askenazi et al have produced a report of a metanalysis of proteomic mass spectrometry studies of brain tissue from Alzheimer's disease patients. The authors have compiled the data from 38 studies, then analysed a number of hypotheses and produced an online resource. The value of this paper is the unification of many different datasets, which will be broadly useful for many subsequent studies.

1. The website is a good and easy to use resource to breakdown specific protein changes seen in regions, stages of pathology, and specific papers

2. Some novelty comes from grouping all the mass spec studies from multiple protocols as well as the addition of studies that have looked specifically at protein changes in isolated lesions rather than bulk preparations.

Points to address:

3. More details about inclusion criteria for bulk proteomics studies: In table 1 or in a supplementary figure it would be good to have a column stating the protocols used in each study. For example, Carlyle et al 2021 was considered as a bulk method study and it uses the SYNPER method, which is method for enriching for synapses, plus TMT labelling mass spec.

4. Due to the low power and number of studies the authors grouped preclinical AD (PCL) and MCI cases. This might be misleading as some preclinical AD cases do not go on to present with AD later in life. They do highlight this in their limitations section but they do mention that the PCL and MCL cohort is limited due to study number and low power.

5. Page 7 Line 19 – "We propose that these protein changes are some of the earliest protein changes in AD, occurring prior to the development of neuropathology and persisting throughout disease progression." – what are they defining as neuropathology in AD? You can already see some neurodegeneration and accumulation of A β /p-Tau in MCI. I think they need to be a bit clearer on what regions are being analysed here to make this conclusion as well.

Minor points:

1. The title will become out of date very soon because it refers to "all known changes". A better title would be something like: "A metanalysis of proteomic studies of Alzheimer's disease brain tissue"

2. Abstract, line 23: Remove "exceptional".

3. Line 31. "proteomic studies were remarkably consistent". I suggest dropping this sentence. Why is it remarkable that 848 proteins out of 5,311 were consistently changed in >5 studies?. Better to keep the hyperbole to a minimum.

4. Introduction line 13. Refers to different tissue fractions, but these are not noted in Table 1. It would be good to have this information in Table 1, if possible.

Reviewer #3 (Remarks to the Author):

In this paper, Askenazi et al compile a meta-analysis of LC-MS/MS studies of post-mortem human brain in Alzheimer's disease into an easily searchable and enlightening public resource.

As with all these endeavours, I could ask for many additional figures that I would love to see, but I do not believe that my own personal interests should delay the publication of this manuscript. This resource will be extremely helpful to the field, and has already changed my way of thinking about a couple of experiments currently happening in my lab.

The supplementary tables are easy to understand, the methods clear and concise, the figures are well plotted, and the purpose of the paper is as a tantalising introduction to a wealth of exciting questions. I can ask these questions myself using this resource.

I can't quite believe, therefore, that the only change I would like to see is a clear index for the supplementary tables, so that one doesn't need to go through the text to find the correct table.

NCOMMS-23-18546-T Reviewer Responses

We thank all the reviewers for taking the time to review our manuscript and their positive and constructive feedback they provided. Our responses to their specific points are included below.

Reviewer #1 (Remarks to the Author)

The manuscript by Askenazi et al. provides a meta-analysis of previous proteomic studies analyzing protein changes in the Alzheimer's disease (AD) by mass spectrometry. The authors compare differentially expressed proteins in preclinical AD, mild cognitive impairment, and advanced AD in different brain regions as well as neuropathologic lesions. Among the main findings are: 1) a high inter-study consistency showing that protein changes in the AD brain occurs in the same direction (up- or downregulation), regardless of brain region, 2) bulk brain tissue and neuropathologies did not show highly similar proteomic changes, indicating that bulk tissue studies cannot be directly used to infer neuropathology changes in AD, 3) identification of protein changes in early vs advanced AD, thus potentially identifying initiating drivers of the disease and drug targets for early AD, 4) almost half of the proteins with the greatest changes in expression in AD are understudied in the AD field with 4 proteins identified that have not been directly linked to AD prior, 5) the earliest affected cellular components were synapses, cytoskeleton, lysosome, and clathrin vesicles, while changes in mitochondrial and neuropathologic lesion proteins were associated with advanced AD, 6) protein changes in vulnerable brain regions in early stages of AD are similar to changes in resistant brain regions in advanced AD, suggesting a temporal wave of progressive dysfunction in different brain regions during AD progression, and 7) the establishment of a searchable database (NeuroPro) that provides a new resource for researchers. Meta-analyses of proteome changes in AD brain have been published before and do, by themselves, not offer a novel concept of analysis. However, when a meta-analysis of other primary as well as meta-analyses studies helps to overcome limitations of smaller study sizes, then there certainly is great value. This manuscript combines 38 published studies with 59 unique comparisons and represents the largest analysis thus far. Askanazi et al. is without a doubt an important study with great significance that lacks major flaws in data analysis, interpretation, and conclusions.

Specific minor points:

1) The introduction nicely outlines tissue fractions from which protein differences have been evaluated, "insoluble, synaptic, membrane, or blood vessel-enriched fractions". A column with this information would be useful for the "Sources" table provided to the NeuroPro website

We have updated the sources table on the NeuroPro website to include the specific tissue fractions that were analysed in each study as well as the lysis and digestion methods that were used. In addition, we have also included this information in Table 1 in the manuscript.

2) Preclinical should be better defined to guide the reader and is not recommended to be shortened to an acronym.

The abbreviation PCL has been removed throughout the manuscript, figures and supplementary data. The following sentence has been included into the methods to clearly define preclinical AD: "In line with the current literature, we defined preclinical AD by presence of amyloid plaques in the absence of cognitive impairment"⁸¹

3) Some proteins such as VGF and SH3GL1 were decreased in AD brain tissue but enriched in neuropathologic lesions. The authors conclude that the "results suggest that VGF and SH3GL1 may have unique roles in AD pathogenesis." Consideration of protocol differences between preparation of bulk tissue and neuropathologic lesions should be discussed, or included as a limitation of interpretation of different protein identifications.

Thank you for highlighting this point – we agree that this is an important limitation. We have revised the paper to now say: "These differences could be due to sample preparation differences between bulk tissue and neuropathological lesion studies, or it may suggest that VGF and SH3GL1 could have unique roles in AD pathogenesis."

4) One limitation is that this study compares differentially-expressed proteins, but does not look at changes in protein modifications (including phosphorylation) which undoubtedly are critical in regulating protein activity, transport etc. Some text in the Discussion regarding this for future studies or text regarding in limitation section.

Excellent point, we very much agree. We have included this text in the discussion to highlight this point: “Additionally, NeuroPro could be expanded to include changes in protein post-translational modifications (e.g. phosphorylation, acetylation, ubiquitination etc.), which are known to be critical disease mediators in AD. Only a small number of proteomic studies examining post-translational modifications in human AD brain tissue have been performed to date, however as studies in this area increase, these would be an excellent future addition to NeuroPro.”

5) The authors address that animal studies are of limited value for incorporation into NeuroPro, but some Discussion of proteomics in iPSCs may be of interest for future studies or as a resource page on NeuroPro.

We had added the following text into the discussion: “Looking forward, NeuroPro will continue to expand as proteomic studies increase in the future. There is potential for inclusion of proteomic studies of disease models (such as iPSCs or animal models), which would be useful for addressing important questions about how closely these models reflect human AD. With regards to iPSCs, it would be important for future studies to compare the proteome of multiple unique patient-derived lines from both AD and controls to reflect the inter-patient variability typically present in human brain tissue studies.”

6) Antibody usage to identify neuropathologic lesions should be included in the Sources table on NeuroPro to help guide the user.

We have updated the sources table to now include this information in the lysis and digestion column on both the sources table on the NeuroPro website and on Table 1 in the manuscript.

7) Figure 1, second panel says 38 studies but then breaks down into 32 and 7. Was there an overlapping study that should be commented upon in the figure legend? The date in Figure 1 is also recommended to be updated to March 2022 as the formatting may not be intuitive to all readers.

We have updated the figure legend to account for this discrepancy (which was an overlapping study) and we have updated the date to March 2022 as suggested.

8) Why do the authors go from supplementary table 2 to supplementary table 8? And then jumped from Supplementary Table 3 to Supplementary Table 10

We have updated these to now be in the order that they appear in the text.

9) Figure 4 is a great summary figure!

Thank you!

10) Generally it is recommended text for “late AD” be updated to “advanced AD”. With LATE-NC representing TDP-43, the use of “late AD” may cause confusion. This is recommended for text and figures. Figure 5A says “late AD studies” under arrow but elsewhere it says “advanced AD.” Supplementary Table 10 says “late AD” – change to “advanced AD.” These should be changed for consistency.

Thank you for highlighting these inconsistencies. We have updated both of these errors.

11) Figure 6B could be more clear in the labelling where it says vulnerability.

We have now included the labels of Low and High above each of the bars to make this figure clearer.

12) Can future proteomic studies be added to the NeuroPro website with either user recommendations or a yearly review of literature?

Yes, the plan is to maintain NeuroPro by a yearly review of the literature.

13) Perhaps update the “Multi-Protein Query” so that it is more user friendly. For example, change it to where the user can click on an experimental protein set to look at novel vs confirmed proteins instead of typing in/deleting protein sets. Or include a reset button to get the experimental protein sets to repopulate

after deletion.

Thank you for this suggestion. We appreciate your ideas to try and improve NeuroPro. We'll continuously work on trying to improve the website based on user feedback such as this.

Reviewer #2 (Remarks to the Author)

Askenazi et al have produced a report of a meta-analysis of proteomic mass spectrometry studies of brain tissue from Alzheimer's disease patients. The authors have compiled the data from 38 studies, then analysed a number of hypotheses and produced an online resource. The value of this paper is the unification of many different datasets, which will be broadly useful for many subsequent studies.

1. The website is a good and easy to use resource to breakdown specific protein changes seen in regions, stages of pathology, and specific papers

2. Some novelty comes from grouping all the mass spec studies from multiple protocols as well as the addition of studies that have looked specifically at protein changes in isolated lesions rather than bulk preparations.

Points to address:

3. More details about inclusion criteria for bulk proteomics studies: In table 1 or in a supplementary figure it would be good to have a column stating the protocols used in each study. For example, Carlyle et al 2021 was considered as a bulk method study and it uses the SYNPER method, which is method for enriching for synapses, plus TMT labelling mass spec.

This is a great suggestion. Based on a similar suggestion also from Reviewer #1 we have updated both Table 1 and the sources table on the NeuroPro website to now include details of the sample lysis and digestion.

4. Due to the low power and number of studies the authors grouped preclinical AD (PCL) and MCI cases. This might be misleading as some preclinical AD cases do not go on to present with AD later in life. They do highlight this in their limitations section but they do mention that the PCL and MCL cohort is limited due to study number and low power.

We agree that this is an important point to highlight. We have now added in the following text in the discussion: "This is additionally important because there is still debate about whether preclinical AD cases are in fact early AD, as they could instead be individuals resilient to AD. Future high-powered proteomic studies directly comparing preclinical AD to MCI are needed to address this important question."

5. Page 7 Line 19 – "We propose that these protein changes are some of the earliest protein changes in AD, occurring prior to the development of neuropathology and persisting throughout disease progression." – what are they defining as neuropathology in AD? You can already see some neurodegeneration and accumulation of A β /p-Tau in MCI. I think they need to be a bit clearer on what regions are being analysed here to make this conclusion as well.

To clarify the resistant brain regions we are referring to here we have updated the first sentence of this paragraph to state the following: "Based on the significant overlap in protein changes in vulnerable and resistant brain regions in advanced AD, we hypothesized that protein changes in resistant brain regions in advanced AD (e.g. sensory cortex, motor cortex, cerebellum) may be the same as those in vulnerable regions in early-stage AD."

This statement is based on the same protein changes being present in resistant brain regions in advanced AD (motor cortex, sensory cortex, cerebellum), as in vulnerable brain regions in early clinical stages of AD (preclinical AD and/or MCI) or vulnerable brain regions in advanced AD. While we do not have the specific data in this study to determine if there was a minor amount of as plaques and/or tangles in resilient brain tissue in individual cases (as relied solely on the information included in these previously published studies), we believe our conclusion is justified based on the lack of significant difference in either APP or tau in these resistant regions, contrasting with increased levels of both APP and tau in vulnerable regions in both early (preclinical AD and/or MCI) and advanced AD. As we do not yet have direct data to back this up, we were deliberately careful in both the results and discussion to present this as a hypothesis rather than a conclusion.

Minor points:

1. The title will become out of date very soon because it refers to “all known changes”. A better title would be something like: “A meta-analysis of proteomic studies of Alzheimer’s disease brain tissue”

Thank you – the title has been updated to one suggested by the editorial team.

2. Abstract, line 23: Remove “exceptional”.

Exceptional has been removed

3. Line 31. “proteomic studies were remarkably consistent”. I suggest dropping this sentence. Why is it remarkable that 848 proteins out of 5,311 were consistently changed in >5 studies?. Better to keep the hyperbole to a minimum.

We have removed this sentence.

4. Introduction line 13. Refers to different tissue fractions, but these are not noted in Table 1. It would be good to have this information in Table 1, if possible.

This information has now been included both in Table 1 and in the sources table on the NeuroPro website.

Reviewer #3 (Remarks to the Author)

In this paper, Askenazi et al compile a meta-analysis of LC-MS/MS studies of post-mortem human brain in Alzheimer's disease into an easily searchable and enlightening public resource.

As with all these endeavours, I could ask for many additional figures that I would love to see, but I do not believe that my own personal interests should delay the publication of this manuscript. This resource will be extremely helpful to the field, and has already changed my way of thinking about a couple of experiments currently happening in my lab.

The supplementary tables are easy to understand, the methods clear and concise, the figures are well plotted, and the purpose of the paper is as a tantalising introduction to a wealth of exciting questions. I can ask these questions myself using this resource.

I can't quite believe, therefore, that the only change I would like to see is a clear index for the supplementary tables, so that one doesn't need to go through the text to find the correct table.

Thank you so much for such a lovely review. We have now included an index of the supplementary tables.